# Gradient Masked Averaging for Federated Learning

**Irene Tenison**[*,1,∇]**, Sai Aravind Sreeramadas**[2,3]**, Vaikkunth Mugunthan**[1]**, Edouard Oyallon**[4]**,
Irina Rish**[2,3]**, and Eugene Belilovsky**[3,5]

[1]*Massachussetts Institute of Technology, MA, USA,* [2]*Universite de Montreal, Quebec, Canada,* [3]*Mila, Quebec AI Institute, Canada,* [4]*Sorbonne University, Paris, France,* [5]*Concordia University, Quebec, Canada* ,
[∇]*itenison@mit.edu*

**Reviewed on OpenReview:** *https: // openreview. net/ forum? id=REAyrhRYAo*

## Abstract

Federated learning (FL) is an emerging paradigm that permits a large number of clients with heterogeneous data to coordinate learning of a unified global model without the need to share data amongst each other. A major challenge in federated learning is the heterogeneity of data across client, which can degrade the performance of standard FL algorithms. Standard FL algorithms involve averaging of model parameters or gradient updates to approximate the global model at the server. However, we argue that in heterogeneous settings, averaging can result in information loss and lead to poor generalization due to the bias induced by dominant client gradients. We hypothesize that to generalize better across non-i.i.d datasets, the algorithms should focus on learning the invariant mechanism that is constant while ignoring spurious mechanisms that differ across clients. Inspired from recent works in Out-of-Distribution generalization, we propose a gradient masked averaging approach for FL as an alternative to the standard averaging of client updates. This aggregation technique for client updates can be adapted as a drop-in replacement in most existing federated algorithms. We perform extensive experiments on multiple FL algorithms with in-distribution, real-world, feature-skewed out-of-distribution, and quantity imbalanced datasets and show that it provides consistent improvements, particularly in the case of heterogeneous clients.

## 1 Introduction

Federated Learning (FL) is a distributed machine learning approach that allows clients with decentralized data to efficiently learn a shared global model without having to share their sensitive datasets McMahan et al. (2017); Kairouz et al. (2021). This enhances privacy as data is neither collected at a central location or cloud nor communicated over any channel. Furthermore, Parcollet et al. (2021) and Qiu et al. (2021) argue that federated learning has a lower carbon footprint than traditional machine learning. A challenge in FL is heterogeneity in the data distributed across clients. The non-i.i.d data distribution degrades the performance of federated learning models Li et al. (2020a); Wang et al. (2019); Zhao et al. (2018). One of the reasons for this is the loss of information on invariances between clients induced by the averaging of model parameters or updates. This is further exacerbated by the multiple local steps taken by each client with the aim of reducing communication rounds, which results in "client drift"Karimireddy et al. (2021). Each client after multiple local steps can progress too far towards minimizing their local objective, which may deviate from that of the global objective.

Many existing FL works attempt to tackle this problem through the lens of optimization, proposing constrained gradient optimization based approaches Karimireddy et al. (2021); Wang et al. (2019); Li et al. (2019) which attempt to maintain a solution close to the global optimum across the federated data. Another branch of work has considered approaches based on knowledge distillation Zhu et al. (2021) to address this problem.

---

*This work was done when Irene Tenison was at Universite de Montreal and Mila, Quebec AI Institute.

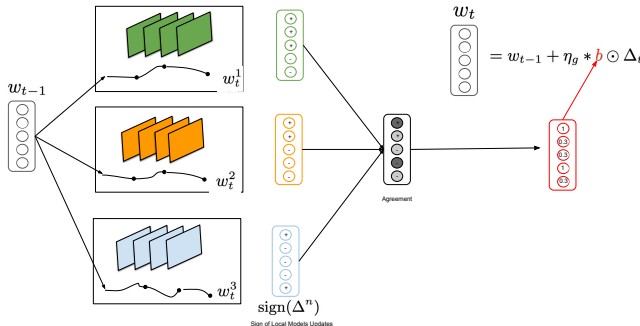

Figure 1: Illustration of Gradient Masked Averaging (GMA). Each client receives a global model and performs multiple iterations to obtain a final update sent to the server. The server uses the sign of each update component to determine an elementwise agreement score. This is then used to adjust the overall global model update, emphasizing components where client's agree on their direction.

On the other hand we can consider the challenge of local training and evaluation on different clients with varying distributions as a distribution shift problem. One can thus consider leveraging many of the recent advances in tackling distribution shifts made in areas such as OOD generalization Arjovsky et al. (2020).

OOD generalization is often formalized using the notion of domains or environments. Under the formalism of Arjovsky et al. (2020) an environment corresponds to a data generating distribution that can be related through underlying (potentially unknown) causal variables to a set of other environments. Different environments can arise during model training and testing, while it is typically assumed that all environments (train and test) share some invariant mechanisms. However, they can have spurious mechanisms that differ across environments Parascandolo et al. (2020); Bühlmann (2018). The concept of environments can be related to the FL setting involving decentralized data by considering each client as producing a set of data generated from a different environment. Clients have invariant mechanisms across them that can lead to robust global models, as well as spurious mechanisms associated to their local data. Leveraging the existing strategies in OOD generalization can thus potentially lead to more robust global model.

Recently Parascandolo et al. (2020) proposed a simple approach to improve generalization in the Out-of-Distribution (OOD) setting. Their approach studied in the centralized setting modifies gradient based optimization by applying a masking operation to individual sample gradients. Identifying this is an approach which can create more robust predictors and tackle the inherent distribution shift problem in FL we propose a new aggregation method called gradient masked averaging (GMA) with the goal of improving generalization across clients and of the global model. GMA operates on client updates from different FL models, determining their component-wise agreement, and subsequently applying a soft-masking operation to perform the final update. It is illustrated in Figure 1.

GMA can be plugged into any FL algorithm as an alternative to naive averaging of model parameters at the server. Intuitively, gradient masking prioritizes gradient components that are aligned with the overall dominant direction across clients while the inconsistent components of the gradient are given lower importance when making the update. Applying this approach leads to improved performance in a variety of training and evaluation settings across a number of standard datasets used for federated learning. Our performance gains are particularly large in cases where the clients are non-iid. Code for our experiments is include in the supplementary materials and will be made available at the time of publication.

**Our Contributions** in this work are as follows:

- We introduce gradient masked averaging as a drop-in robust alternative to naive averaging of parameters.

- We empirically show that applying gradient masking to any FL algorithm consistently improves performance of the algorithm. This improvement was observed on in-distribution evaluations and its various settings, quantity skewed datasets, real world evaluations, and more complex out-of-distribution evaluation datasets.

## 2 Related Work

**Federated Learning**    In FedAVG McMahan et al. (2017) for each communication round, all randomly selected $B$ fraction of clients (called participating clients) perform $E$ local steps of gradient descent with their local datasets. The model parameters from participating clients are averaged at the server to obtain the global model. It is equivalent to FedSGD McMahan et al. (2017) when $E = 1$ and each client performs stochastic gradient descent. Multiple local steps help minimize communication costs, which is a major bottleneck in FL. Quantization methods Reisizadeh et al. (2020) and gradient descent acceleration Yuan & Ma (2021) methods have been proposed to reduce communication overhead. Convergence of FedAVG under i.i.d settings have been analyzed widely Stich (2019); Yu et al. (2018); Wang & Joshi (2019). The convergence rate of FedAVG worsens with increasing heterogeneity among client datasets and this has been analyzed by several works Li et al. (2020a); Wang et al. (2019). Multiple variations of FedAVG have been proposed to improve convergence in non-i.i.d data distribution settings, including adding regularization to the client objective Li et al. (2020a), normalized averaging of model parameters Wang et al. (2020b), introducing server momentum Hsu et al. (2019) or adaptive optimizers at the server Reddi et al. (2021), and introducing control variates Karimireddy et al. (2021). Generalization in federated learning has been explored in terms of out-of-sample and participation gaps in Yuan et al. (2021). Francis et al. (2021) explores adaptations of IRM to exploit invariance in federated learning, Gupta et al. (2022) explores the same from a game theoretic perspective, and Zhang et al. (2021) employs federated adversarial training for the same. PFNM Yurochkin et al. (2019) and FedMAWang et al. (2020a) addresses the problems due to permutation variances in the neural networks. Algorithms like PerFedAVG Fallah et al. (2020), Ditto Li et al. (2021b), FedBABU Oh et al. (2021), FedSHIBU Raj et al. (2022) focus on client personalization.

**Out of Distribution Generalization**    In traditional machine learning, a model is evaluated based on its test performance on an unseen dataset drawn i.i.d from the train data distribution. However, this assumption may not be true in real-world datasets and many supervised learning models do not perform well on related but non-i.i.d test datasets. This problem is often referred to as the out-of-distribution generalization or the closely related domain generalization problem Koyama & Yamaguchi (2021); Ahuja et al. (2020). This problem has been addressed in several works like Invariant Risk Minimization(IRM) Arjovsky et al. (2020), Risk Extrapolation(REx) Krueger et al. (2021), and Gradient Starvation Pezeshki et al. (2021). These approaches typically focus on introducing penalties that learn invariant representations in a setting with known variations in the data (corresponding to environments). However, this idea cannot be easily ported to a FL setting as the clients performing the optimization steps would require access to the data of other clients. On the other hand Parascandolo et al. (2020) proposed a gradient agreement method based on gradient directions to learn features that agree across environments. This was extended by Shahtalebi et al. (2021) to include gradient magnitude. In this paper we focus on this class of gradient agreement methods. Distinct from the prior work, which considers the case of individual samples and single global updates, we adapt this approach to a federated setting, where each client produces an aggregate update based on multiple gradient iterations.

## 3 Methods

In the following section we introduce the notations used, review standard federated aggregation, and then introduce gradient masked averaging for global model approximation.

### 3.1 Federated Aggregation

Fix $\mathcal{D}$ a training set that we assume splitted across $N$ clients in $\{\mathcal{D}^n\}_{n \leq N}$ local datasets of size $s_n$. The objective of the clients is to collectively learn a function via supervision, $f : \mathbb{R}^{\bar{d}} \to \mathbb{R}$ that, in our case, corresponds to a neural network model with parameters, $w \in \mathbb{R}^d$. For this, we define the global objective function in terms of the local objective, $F_n$, of each client as shown below, assuming that $\mathbb{E}_{z^n \sim \mathcal{D}^n}[f_n(w; z^n)] = F_n(w)$, where $f_n$

is the clients objective function. In our setting, we are thus interested in minimizing:

$$\min_w f(w) = \min_w \sum_{n=1}^{N} F_n(w),$$

When using the local data, with each client having $s_n$ samples, we focus on minimizing the empirical objective function, given by:

$$\min_w \sum_{n=1}^{N} \frac{1}{s_n} \sum_{z^n \in \mathcal{D}^n} f_n(w; z^n).$$

For this, we consider a standard first order minimization scheme. At each communication round, $t$, and for a total of $T$ iterations, the parameters of the global model, $w_t$, are sent to each of the $N$ participating clients which perform multiple local gradient steps to obtain a local update, $\Delta_t^n$, corresponding to the difference between the clients model after multiple updates and $w_t$. In most FL algorithms the global model update at $t^{th}$ communication round is then obtained as

$$w_{t+1} = w_t - \eta_g \Delta_t \quad \text{where} \quad \Delta_t = \frac{1}{|N|} \sum_{n \in N} \Delta_t^n, \tag{1}$$

$\eta_g$ is the global learning rate and $\Delta_t$ is the update or "pseudo-gradient" at the $t^{th}$ global communication round obtained by aggregating the updates from the participating clients $\{\Delta_t^n\}_{n \leq N}$. In this context, for the local learning rate client $\eta_l$, we obtain a local update during $K$ iterations, via local parameters $\{w_{t,k}^n\}_{k \leq K}$ and batches $z_{t,k}^n$ given by:

$$\Delta_t^n = -\eta_l \frac{s_n}{\sum_{n \leq N} s_n} \sum_{k=1}^{K} \nabla f_n(w_{t,k}^n; z_{t,k}^n). \tag{2}$$

In the case of a single gradient step at each client, the update $\Delta_t$, corresponds to the gradient of the global objective. Each client has a different data distribution and thus different loss surface. Parascandolo et al. (2020) shows that averaging of gradients across environments leads to poor consistency of solutions, and reduced generalization, particularly to unseen environments. Indeed naive averaging of parameters fails to capture the consistencies in the loss landscapes due to the bias that may be induced by dominant features in the environments as explained by Shahtalebi et al. (2021). This is further exacerbated in real world federated settings as there are multiple possible scenarios where some clients dominate over others.

## 3.2 Gradient Masked Aggregation

Parascandolo et al. (2020) highlights several issues associated with the standard arithmetic mean, which they equate to a logical OR. They propose to use an analog of the logical AND operation resulting in taking geometric mean of sample gradients. The gradient components that are "inconsistent" in sign across environments are set to 0. Specifically they construct a binary vector, $m_\tau$ based on the agreement of gradient components among environments. The components $j$ of the mask, $m_\tau$ is computed as $[m_\tau]_j = \mathbb{1}\|\frac{1}{|N|} \sum_{e \in \eta} \text{sign}([\nabla L_e]_j) \geq \tau\|$. Here $\nabla L_e$ is the loss gradient for environment $e$, and $\tau \in [0, 1]$ is a hyperparameter.

Direct application of this idea to the FL setting is however challenging. Parascandolo et al. (2020) applies the rule assuming each sample represents an environment, whereas each client more naturally corresponds to the environment in FL. Furthermore, they show that this can lead to a slower convergence rate in practice as too many components can be masked at each iteration. The same was observed in a federated setting as shown in Appendix. This would be impractical in the federated setting as we would not want to sacrifice convergence speed for generalization. In the federated setting, we propose a variant of this mask that doesn't sacrifice convergence speed while retaining some of the improved generalization properties. Specifically, we propose to use masking at the aggregation stage of standard FL, with a mask computed based on each client update (which arises from multiple local gradient steps). The mask is calculated based on sign agreement among client updates $\Delta_t$ and applied on the global model update $\Delta_t^n$. This masking controls the parameter update based on the agreement of direction among the gradients across clients or environments. To provide rapid convergence, we apply a soft masking procedure instead of the hard binary mask.

**Algorithm 1** Gradient Masked FedAVG McMahan et al. (2017)

---

**Server Executes:**

    Initialize $w_0 \in \mathbb{R}^d$ randomly
    **for** each server epoch, $t = 1, 2, 3, ..., T$ **do**
        Choose C clients at random
        **for** each client $n$ in C **do**
            $w_t^n = \text{ClientUpdate}(w_{t-1}, n)$
            $\Delta_t^n = \frac{s_n}{\sum_{k \in C} s_k}(w_t^n - w_{t-1})$
        **end for**
        $\Delta_t = \sum_{n \in C} \Delta_t^n$
        $m_t = \tilde{m}_\tau(\{\Delta_t^n\}_{n \in C})$
        $w_t = w_{t-1} + \eta_g * m_t \odot \Delta_t$
    **end for**

**ClientUpdate(w, n):**

    Initialize $w_0 = w$
    **for** each local client iteration, $k = 0, 1, 2, 3, .., K-1$
    **do**
        Sample $z \sim \mathcal{D}^n$
        $g_k = \nabla_w F_n(w_k; z)$
        $w_{k+1} = w_k - \eta_l \ g_k$
    **end for**
    **return** $w_K$ to server

---

We define an agreement score, $A_j \in (0, 1]$, given as a function of all client updates, and the mask $\tilde{m}_\tau$ is defined element-wise,

$$[\tilde{m}_\tau]_j = 1 \text{ if } A_j \geq \tau \text{ else } A_j; \quad A = \left| \frac{1}{|N|} \sum_{n \leq N} \text{sign}(\Delta^n) \right| \tag{3}$$

The global model update is given by $\tilde{m}_\tau(\{\Delta^n\}) \odot \Delta$. This ensures that the updates to the global model are with respect to their agreement across clients. When the agreement across clients is greater than the hyperparameter $\tau$, it would be assigned 1 and when the agreement is lesser than $\tau$, the mask value would be equivalent to the agreement score. This real mask ensures that each parameter updates, but the magnitude is adjusted to be proportional to the agreement across clients. We observe empirically in Appendix that the fraction of clients whose magnitude gets adjusted as mentioned above is correlated to the heterogeneity in the distribution of data across clients. The method can be applied to any base FL optimizer, in Algorithm 1 we detail its implementation for FedAVG.

## 4 Method Analysis

We now analyze the convergence properties of the gradient masking, focusing on the case of FedAVG and gradient masked aggregation. To obtain a standard convergence result our algorithm requires to lower-bound on the norm of the masked gradients, requiring additional assumptions. We propose a simplification, which allows to obtains similar convergence guarantees as Reddi et al. (2021): we assume that the binary masks follow independent Bernoulli distributions. Also, note that we propose our proofs in the context of FedAVG rather than FedADAM, making the analysis more straightforward. Following Reddi et al. (2021), we make the following standard assumptions.

**Assumption 4.1** (Lipschitz gradient)**.** We assume that each client objective has Lipschitz gradient with constant $L$, meaning that there exists $L > 0, \forall n \leq N, \forall w, v \in \mathbb{R}^d, \|\nabla F_n(w) - \nabla F_n(v)\| \leq L\|w - v\|$

**Assumption 4.2** (Bounded gradients)**.** We assume that each client has a bounded gradient by $G$, leading to: $\exists G > 0, \forall n \leq N, \forall w \in \mathbb{R}^d, \|\nabla F_n(w)\| \leq G$.

**Assumption 4.3** (Finite variance)**.** We assume a global bound on the variance of the gradient estimate of each individual client, meaning that: $\exists \sigma > 0, \forall n \leq N, \forall w \in \mathbb{R}^d, j \leq d, \mathbb{E}_z[\nabla F_n(w) - \nabla f_n(w; z)]_j^2 \leq \sigma_{l,j}^2$.

**Assumption 4.4** (Global variance)**.** We assume a global bound on the variance of the gradient estimate of each individual client, meaning that: $\forall j, \exists \sigma_{g,j} > 0, \forall n, \forall w, \frac{1}{N} \sum_{n=1}^N (\nabla F_n(w) - \nabla f(w))_j^2 \leq \sigma_{g,j}^2$.

The next assumption is core and allows to work with random masks $m_t^j \in \{0, 1\}$ at step $t$, assuming every coordinates could be unmasked with a non zero probability..

**Assumption 4.5** (Random mask.)**.** We assume that $\inf_{j \leq d} \mathbb{E}[m_t^j] \geq \alpha > 0$ and its sampling is independent from $\nabla f_n(w_t; z_t^n)$.

We begin by this first Lemma, which allows to work with weighted agregations of the local gradients:

**Lemma 4.6** (Adaptation from Reddi et al, using the notations of Reddi et al. (2021))**.** *If $\{w_{n,k}^t\}_{k \leq K}$ are the $K$ local iterations of the $n$-th machine at time $t$, we have for any $\sum_n \lambda_n = 1$, $\lambda_i \geq 0$:*

$$\sum_{n=1}^N \lambda_i \mathbb{E}[\|w_{n,k}^t - w_t\|^2] \leq 5K\eta_l^2 \mathbb{E} \sum_{j=1}^d (\sigma_{l,j}^2 + 2K\sigma_{g,j}^2) + 30K^2\eta_l^2 \mathbb{E}[\|\nabla f(w_t)\|^2]$$

*Proof.* As done in Appendix A, Lemma 3 of Reddi et al. (2021), we will prove the result by induction, and we use the same notations. First, we use the inequality proved by Lemma 3 of Reddi et al. (2021):

$$\mathbb{E}\|w_{n,k}^t - w_t\|^2 \leq (1 + \frac{1}{2K-1} + 6K\eta_l^2 L^2)\mathbb{E}[\|w_{n,k-1}^t - w_t\|^2] + \eta_l^2 \mathbb{E}\sum_{j=1}\sigma_{l,j}^2 + 6K\mathbb{E}[\|\eta_l(\nabla F_n(w_t) - \nabla f(w_t))\|^2$$
$$+ 6K\eta_l^2\mathbb{E}\|\nabla f(w_t)\|^2]$$

Next, we note that by weighted averaging that:

$$\sum_{n=1}^N \lambda_i\mathbb{E}\|w_{n,k}^t - w_t\|^2 \leq (1 + \frac{1}{2K-1} + 6K\eta_l^2 L^2)\sum_{n=1}^N \lambda_i\mathbb{E}[\|(w_{n,k-1}^t - w_t)\|^2] + \eta_l^2\mathbb{E}\sum_{j=1}\sigma_{l,j}^2$$
$$+ 6K\sum_{n=1}^N \lambda_i\mathbb{E}[\|\eta_l(\nabla F_n(w_t) - \nabla f(w_t))\|^2 + 6K\eta_l^2\mathbb{E}\|\nabla f(w_t)\|^2]$$

Now, using our assumption, we get:

$$\sum_{i=1}^m \lambda_i\mathbb{E}[\|(\nabla F_i(w_t) - \nabla f(w_t))\|^2 \leq \sum_{i=1}^m \lambda_i\sum_{j=1}^d \sigma_{g,j}^2 = \sum_{j=1}^d \sigma_{g,j}^2 \tag{4}$$

The rest of the proof follows directly as in Reddi et al. (2021) as it relies purely on $\phi_k \triangleq \sum_{n=1}^N \lambda_n\mathbb{E}\|w_{n,k}^t - w_t\|^2$. □

Our proofs will rely on the following Lemma, where the constant $C$ is specified in Lemma A4.6 given in the Appendix. We emphasize that the logic of this proof and the constants are identical to Reddi et al. (2021).

**Lemma 4.7** (Bounded drift from client update, Adapted from Appendix A, Lemma 3 of Reddi et al. (2021)). *Given Assumptions 4.1, 4.2, 4.3, 4.4, there exists $C > 0$ such that for any time step $t$ and $w_t, \Delta_t$ obtained from Alg. 1, if $\eta_l \leq \frac{1}{8LK}$:*

$$\mathbb{E}[\|\Delta_t + K\eta_l\nabla f(w_t)\|^2] \leq 5K^2(\eta_l^2 + L^2\eta_l^4)\mathbb{E}\sum_{j=1}^d (\sigma_{l,j}^2 + 3K\sigma_{g,j}^2) + 30L^2K^3\eta_l^4\mathbb{E}[\|\nabla f(w_t)\|^2]$$

*and*

$$\|\mathbb{E}[\Delta_t + K\eta_l\nabla f(w_t)]\|^2 \leq 5^2KL^2\eta_l^4\mathbb{E}\sum_{j=1}^d (\sigma_{l,j}^2 + 3K\sigma_{g,j}^2) + 30L^2K^3\eta_l^4\mathbb{E}[\|\nabla f(w_t)\|^2]$$

Its proof can be found in the appendix. The next proposition derives an upper bound on the model gradient following results from Reddi et al. (2021), and relates the speed of convergence to $\alpha$.

**Proposition 4.8** (Convergence analysis). *Given Assumptions 4.1, 4.2, 4.3, if $\eta_g = \mathcal{O}(1)$ then there exists $\eta_l = \mathcal{O}(\frac{\alpha}{KL})$, such that one has the following rate over the masked gradients given by the FedAVG algorithm :*

$$\frac{1}{T}\sum_{t=0}^{T-1} \|\nabla f(w_t)\|^2 \leq \mathcal{O}(\frac{1}{\sqrt{\alpha^2 T}} + \frac{1}{\alpha^2 T})$$

*Proof.* We follow the approach of Bottou (2010) for obtaining optimal non-convex bounds. Each $F_n$ is $L$-smooth, thus so is $f$ by averaging, thus, by definition using Alg. 1:

$$f(w_{t+1}) \leq f(w_t) + \langle \nabla f(w_t), w_{t+1} - w_t\rangle + \frac{L}{2}\|w_{t+1} - w_t\|^2$$
$$= f(w_t) + \eta_g\langle \nabla f(w_t), m_t \odot \Delta_t\rangle + \frac{L}{2}\eta_g^2\|m_t \odot \Delta_t\|^2$$
$$\leq f(w_t) + \eta_g\langle \nabla f(w_t), m_t \odot \Delta_t\rangle + \frac{L}{2}\eta_g^2\|\Delta_t\|^2$$

Next, we observe on one side that:

$$\mathbb{E}_{z_{n,t}}\|\Delta_t\|^2 \leq 2\mathbb{E}_{z_{n,t}}\|\Delta_t + K\eta_l\nabla f(w_t)\|^2 + 2K^2\eta_l^2\|\nabla f(w_t)\|^2$$

On the other side, we note that:

$$\eta_g\langle\nabla f(w_t), m_t \odot \Delta_t\rangle = \eta_g\langle\nabla f(w_t), -K\eta_l(m_t \odot \nabla f(w_t)) + K\eta_l(m_t \odot \nabla f(w_t)) + m_t \odot \Delta_t\rangle \quad (5)$$

$$= -\eta_g K\eta_l\|m_t \odot \nabla f(w_t)\|^2 + \eta_g\langle\nabla f(w_t), m_t \odot (K\eta_l\nabla f(w_t) + \Delta_t)\rangle \quad (6)$$

$$= -\eta_g K\eta_l\|m_t \odot \nabla f(w_t)\|^2 + \eta_g\langle m_t \odot \nabla f(w_t), K\eta_l\nabla f(w_t) + \Delta_t\rangle \quad (7)$$

$$\quad (8)$$

Then, taking the expectation over data $z_{n,t,k}$ and an application of Cauchy-Schwartz leads to:

$$\mathbb{E}_{z_{n,t}}\eta_g\langle\nabla f(w_t), m_t \odot \Delta_t\rangle \leq (-\frac{\eta_g\eta_l}{2}K + \frac{\eta_g\eta_l}{2})\|m_t \odot \nabla f(w_t)\|^2 + \frac{\eta_g}{2\eta_l}\|\mathbb{E}\Delta_t + K\eta_l\nabla f(w_t)\|^2$$

Now, we note that taking the expectation over the mask gives $\alpha\|\nabla f(w_t)\|^2 \leq \mathbb{E}\|m_t \odot \nabla f(w_t)\|^2$. Combining and summing for $t = 0, ..., T-1$, we get:

$$0 \leq f(w_0) - f(w^*) + \sum_{t=0}^{T-1}(K^2L\eta_l^2\eta_g^2 - \alpha(K-1)\eta_l\frac{\eta_g}{2})\|\nabla f(w_t)\|^2 + (L\eta_g^2\mathbb{E}[\|\Delta_t + \nabla f(w_t)\|^2] + \frac{\eta_g}{2\eta_l}\|\mathbb{E}\Delta_t + \nabla f(w_t)\|^2)$$

For the sake of simplicity, we introduce $A = 5K^2\mathbb{E}\sum_{j=1}^d(\sigma_{l,j}^2 + 3K\sigma_{g,j}^2)$ and $B = 30K^3$, so that using Lemma 4.7:

$$0 \leq f(w_0) - f(w^*) + \sum_{t=0}^{T-1}(K^2L\eta_l^2\eta_g^2 - \alpha(K-1)\eta_l\frac{\eta_g}{2})\|\nabla f(w_t)\|^2 + L\eta_g^2(A(L^2\eta_l^4 + \eta_l^2) + (A+B)\frac{\eta_g}{2\eta_l}L^2\eta_l^4 + 2B\eta_l^4L^2\mathbb{E}[\|\nabla f(w_t)\|^2])$$

which also writes:

$$0 \leq f(w_0) - f(w^*) + \sum_{t=0}^{T-1}(K^2L\eta_l^2\eta_g^2 + 2B\eta_l^4\eta_g^2L^3 + \frac{B}{2}\eta_gL^2\eta_l^3 - \alpha(K-1)\eta_l\frac{\eta_g}{2})\|\nabla f(w_t)\|^2 + L\eta_g^2A(L^2\eta_l^4 + \eta_l^2) + \frac{\eta_g}{2}(A+B)L^2\eta_l^3$$

Here, let's pick $\eta_g = 1$, leading to, for $0 < \eta_l < 0.001\frac{\alpha}{KL}$ (where we clearly have suboptimal constants):

$$0 \leq f(w_0) - f(w^*) + \sum_{t=0}^{T-1}-\alpha\eta_l\frac{K}{8}\|\nabla f(w_t)\|^2 + 10KL\eta_l^2$$

Reorganizing the term, this writes fir $0 < \eta_l < 0.001\frac{\alpha}{KL}$ and $\sigma^2$ an upper bound over the sum of the variances:

$$\frac{1}{T}\sum_{t=0}^{T-1}\|\nabla f(w_t)\|^2 \leq 8\frac{f(w_0) - f(w^*)}{\alpha\eta_lKT} + \frac{80L\eta_l\sigma^2}{\alpha}$$

Now, we invoke Lemma 17 of Koloskova et al. (2020) which states that there exists $\eta_l^* < 0.001\frac{\alpha}{KL}$ such that there is an absolute $C > 0$ satisfying:

$$\frac{1}{T}\sum_{t=0}^{T-1}\|\nabla f(w_t)\|^2 \leq C(\frac{1}{\alpha}\sqrt{L\sigma^2\frac{f(w_0) - f(w^*)}{KT}} + \frac{L(f(w_0) - f(w^*))}{\alpha^2T})$$

and this leads to the final conclusion.

$\square$

While Assumption 4.5 may not always hold in practice, it allows us to explore the highly non-convex nature of masks given by Eq. 3. This theoretical analysis serves as a sanity check, providing insights that the process of masking is unlikely to significantly impede convergence towards a particular local minimum. However, it is important to note that this theorem alone does not explain the observed strong performance. Instead, it indicates that there are no inherent obstacles preventing the algorithm from converging to a local minimum. Therefore, this simplification of the problem setting serves as a valuable starting point for further exploration and refinement of the masking approach, allowing for the investigation of more realistic and complex scenarios.

## 5 Experiments

In this section, we show empirically that the proposed GMA tends to outperform standard aggregation (AVG), converging at similar or better than standard aggregation, while enhancing the global model generalization. We observe this for multiple FL algorithms with respect to multiple datasets and data distributions (i.i.d and non-i.i.d).

**Implementation** We conduct experiments on gradient masked and naive versions of non-adaptive federated optimizers like FedAVG McMahan et al. (2017), FedProx Li et al. (2020b) and adaptive federated optimizers like FedAdam and FedYogi Reddi et al. (2021) across a variety of datasets. Our experiment include label distribution skew, feature distribution skew, quantity skew, real-world data distribution, and mixed (label and feature) skew. The code is available at https://github.com/arvi797/FL. Details of the datasets explored and the respective skews induced is summarised in Table 1. The reported performances in Tables 2 and 3 are average accuracies of 4 independent runs of the model on a test dataset of the corresponding dataset, data skew, and algorithm.

### 5.1 In-Distribution Evaluation

This is the most widely considered setting in the FL literature. The global model is evaluated on a test dataset sampled from data at all clients irrespective of the data distribution across clients. This test dataset is a representation of all participating clients. Table 2 shows the test performance of the algorithms and their GMA versions on a variety of datasets and data splits. It was observed that with the robust features of GMA, the algorithm is capable of outperforming naive averaging. The difference in improvement is more significant when the data distribution is non-i.i.d. The major reason for this is that gradient masking is capable of focusing on learning the invariances even under increased spuriousness of non-i.i.d data distribution.

**Hyperparameters** An SGD optimizer with a momentum ($\rho = 0.9$) and cross-entropy loss was used to train each client before aggregation at the server in all our experiments. The momentum parameters of adaptive federated optimizers are fixed at $\beta_1 = 0.9$ and $\beta_2 = 0.99$ as per Reddi et al. (2021). For each of the considered algorithms we tune the local client model learning rates, global model learning rates, tau, and number of local epochs (CIFAR and TinyImageNet) to consider the best performances of the algorithms. More details on the hyperparameter tuning is given in Appendix.

| Dataset | Skew | # of classes | # classes in label skew | total # clients | sampled # clients | Model | # local epochs |
|---|---|---|---|---|---|---|---|
| MNIST | label | 10 | 2 | 500, 100, 10 | 10, 10, 10 | LeNet | 1 |
| FMNIST | label | 10 | 2 | 500, 100, 10 | 10, 10, 10 | LeNet | 1 |
| CIFAR10 | label, quantity | 10 | 2, - | 500, 100, 10 | 10, 10, 10 | ResNet18 | 3 |
| CIFAR100 | label | 100 | 10 | 500, 100, 10 | 10, 10, 10 | ResNet18 | 3 |
| TinyImageNet | label | 200 | 20 | 500, 100, 10 | 10, 10, 10 | ResNet18 | 12 |
| FEMNIST | real | 10 | - | 3550, 3550 | 35, 350 | LeNet | 5 |
| FedCMNIST | feature, mixed | 10 | -, 2 | 500, 100, 10 | 10, 10, 10 | LeNet | 1 |

Table 1: Datasets used, heterogeneity induced, and other parameters corresponding to the dataset.

| Dataset | Non-IID | Number of clients | FedAVG AVG | GMA | FedProx AVG | GMA | FedAdam AVG | GMA | FedYogi AVG | GMA |
|---|---|---|---|---|---|---|---|---|---|---|
| MNIST | Yes | 500 | 96.24±0.03 | **97.36±0.02** | 95.80±0.01 | **96.55±0.03** | 97.75±0.05 | **98.48±0.05** | 98.21±0.06 | **98.38±0.03** |
| | | 100 | 98.6±0.02 | **98.75±0.02** | 97.76±0.02 | **98.64± 0.00** | 98.88±0.04 | **98.94±0.03** | 98.58±0.01 | 98.82±0.07 |
| | | 10 | 99.07±0.02 | **99.09±0.02** | 98.91±0.02 | 99.08±0.03 | **98.51±0.05** | 98.44±0.05 | 97.54±0.07 | **98.74±0.01** |
| | No | 500 | **97.14±0.02** | 96.85±0.01 | **97.27±0.03** | 96.53±0.03 | 98.02±0.01 | **98.8±0.02** | 97.49±0.00 | **98.18±0.01** |
| | | 100 | 98.28±0.02 | **98.65±0.02** | 98.72±0.02 | **98.79±0.03** | **98.94±0.02** | 98.72±0.01 | 98.75±0.00 | **98.82±0.01** |
| | | 10 | 99.11±0.02 | **99.17±0.03** | 99.13±0.02 | **99.16±0.02** | 98.71±0.02 | **98.77±0.02** | 98.68±0.01 | **98.75±0.00** |
| FMNIST | Yes | 500 | 78.02±0.08 | **78.82±0.11** | 76.21±0.12 | **77.77±0.12** | 78.80±0.03 | **81.28±0.02** | 78.66±0.11 | 80.58±0.14 |
| | | 100 | 85.58±0.04 | **86.27±0.05** | 85.14±0.12 | **85.78±0.08** | 85.91±0.04 | **87.18±0.01** | 85.32±0.09 | 86.4±0.14 |
| | | 10 | 87.43±0.05 | **88.37±0.07** | 88.25±0.14 | **88.55±0.12** | 87.26±0.07 | **87.50±0.02** | 87.39±0.12 | **88.6±0.11** |
| | No | 500 | 83.07±0.06 | **83.94±0.12** | **83.47±0.08** | 82.77±0.12 | 85.44±0.28 | **86.73±0.26** | 85.55±0.08 | 86.42±0.16 |
| | | 100 | 88.03±0.03 | **88.43±0.08** | 87.05±0.02 | **87.46±0.04** | 88.66±0.18 | 89.16±0.22 | 88.63±0.08 | **89.7±0.22** |
| | | 10 | 89.35±0.02 | **89.65±0.03** | 89.92±0.12 | **90.12±0.03** | 88.66±0.22 | 88.86±0.18 | 88.85±0.18 | 89.85±0.21 |
| CIFAR 10 | Yes | 500 | 73.38±0.46 | **74.44±0.22** | 72.21±0.22 | **73.95±0.36** | 76.41±0.08 | **77.12±0.12** | 76.13±0.08 | **77.56±0.04** |
| | | 100 | 85.55±0.41 | **85.62±0.38** | 84.11±0.36 | **85.37±0.22** | 84.43±0.08 | **85.52±0.08** | 84.62±0.11 | **85.95±0.12** |
| | | 10 | 84.06±0.31 | **85.21±±0.26** | 84.65±0.41 | **85.84±0.38** | 84.63±0.04 | **85.77±0.03** | 84.24±0.04 | **85.32±0.04** |
| | No | 500 | 75.81±0.15 | **76.93±0.11** | 74.43±0.08 | 74.19±0.08 | 77.07±0.12 | **77.57±0.12** | 77.87±0.08 | **78.11±0.06** |
| | | 100 | **80.42±0.14** | 80.07±0.16 | **79.84±0.09** | 79.6±0.12 | 84.18±0.08 | **85.59±0.07** | 84.13±0.11 | **85.27±0.09** |
| | | 10 | 86.47±0.04 | **87.38±0.03** | **87.04±0.06** | 86.89±0.14 | 87.04±0.12 | **87.28±0.16** | 87.06±0.11 | **87.17±0.14** |
| CIFAR 100 | Yes | 500 | 42.68±0.55 | **44.04±0.61** | 38.45±0.64 | **40.74±0.33** | 47.57±0.42 | **50.37±0.42** | 46.27±0.62 | 48.89±0.46 |
| | | 100 | 51.65±0.55 | **53.83±0.42** | 49.81±0.44 | **50.76±0.21** | 53.4±0.52 | **56.68±0.56** | 54.64±0.43 | **56.88±0.38** |
| | | 10 | 50.88±0.22 | **52.03±0.18** | 51.12±0.36 | **51.76±0.36** | 54.66±0.28 | **56.93±0.26** | 54.55±0.22 | **56.73±0.28** |
| | No | 500 | 44.36±0.28 | **46.13±0.28** | 38.6±0.21 | **39.55±0.23** | 44.01±0.14 | **46.38±0.18** | 41.12±0.12 | 44.63±0.08 |
| | | 100 | 56.88±0.05 | **57.19±0.03** | 53.31±0.12 | **54.13±0.08** | 51.33±0.11 | **51.65±0.08** | 51.19±0.08 | **51.41±0.12** |
| | | 10 | 56.80±0.02 | **57.21±0.03** | 58.16±0.04 | **58.21±0.04** | 57.41±0.08 | **57.77±0.12** | **55.93±0.06** | 55.91±0.08 |
| Tiny Image Net | Yes | 500 | 20.24±0.04 | **21.42±0.03** | 21.18±0.08 | **22.36±0.06** | 23.65±0.12 | **25.92±0.06** | 23.81±0.13 | **25.85±0.18** |
| | | 100 | 25.29±0.08 | **26.94±0.08** | **25.64±0.03** | 25.11±0.11 | 25.67±0.17 | **27.42±0.13** | 26.32±0.06 | **28.83±0.12** |
| | | 10 | 25.17±0.08 | **26.17±0.08** | 26.32±0.04 | **26.71±0.04** | 27.41±0.08 | **28.11±0.12** | 27.12±0.08 | **27.88±0.12** |
| | No | 500 | 24.12±0.18 | **24.87±0.12** | **24.73±0.15** | 24.55±0.32 | 21.17±0.12 | **23.73±0.14** | 23.86±0.16 | **25.32±0.11** |
| | | 100 | **26.71±0.12** | 27.00±0.16 | **26.59±0.15** | 26.55±0.35 | 26.99±0.27 | **29.07±0.19** | 27.86±0.17 | **29.32±0.12** |
| | | 10 | **30.66±0.02** | 30.45±0.03 | 29.04±0.04 | **29.83±0.04** | 28.98±0.04 | **32.02±0.08** | 27.75±0.06 | **30.78±0.05** |

Table 2: In-Distribution evaluations in both iid and Non-iid settings on MNIST, FMNIST, CIFAR 10, CIFAR 100, and TinyImageNet distributed across 500, 100, and 10 clients. In all cases, 10 clients are randomly sample fr participation in each round. Average best test performance of the algorithms and their GMA versions are reported below. In most settings and FL algorithms GMA provides substantial advantages. Improvements are more substantial in Non-iid settings. The best result in each setting across all FL algorithms is shown in red (if GMA) and blue (if AVG). Except in one case GMA version of the algorithm provides the overall best performance.

## 5.2 Real-World Evaluation

In the practical federated setting, the data across clients is heterogeneous and the clients which deploy the global model (including test clients, non-participating clients, and new clients in the federated network) can have data distribution different from that at any train clients. This can be simulated by using a realistic federated data characterised by a feature distribution skew unique to each client including the test clients. Specifically, we use train data from the same domain distributed across clients such that each client has data corresponding to one user (or a set of users) unique to the client. The test data consists of data from the same domain as the train dataset distributed across clients but from one user (or a set of users) not included in the set of train clients. The test clients do not participate in any training rounds. That is, it is out-of-distribution to the train data. A similar feature skew is provided by Federated EMNIST Caldas et al. (2019) where the data points have a user identifier. The test performance of the algorithms and their gradient masked alternatives are given in Table 3. We observe that gradient masking outperforms naive averaging. In the next section we consider a more complex feature skewed out-of-distribution to further evaluate GMA.

| Dataset | Mixed Non-IID | # clients, # sampled | FedAVG AVG | FedAVG GMA | FedProx AVG | FedProx GMA | FedAdam AVG | FedAdam GMA | FedYogi AVG | FedYogi GMA |
|---|---|---|---|---|---|---|---|---|---|---|---|
| **Real-World Evaluation** | | | | | | | | | | |
| FEMNIST | - | 3000, 30 | 94.28±0.02 | **95.57±0.05** | 94.75±0.00 | **95.01±0.01** | 90.28± 0.03 | **93.02± 0.02** | 91.22± 0.02 | **92.79± 0.02** |
| **Out-of-Distribution Evaluation** | | | | | | | | | | |
| | Yes | 500, 10 | 84.48±0.27 | **86.22± 0.22** | 83.15±0.17 | **86.25±0.28** | 84.22±0.42 | **87.16±0.37** | 84.06±0.36 | **87.05±0.44** |
| Fed-CMNIST | | 100, 10 | 85.62± 0.28 | **88.32± 0.32** | 83.3±0.33 | **86.92± 0.28** | 84.19± 0.42 | **86.26± 0.38** | 83.32±0.48 | **84.79± 0.22** |
| | | 10, 10 | 86.77± 0.43 | **89.17± 0.38** | 86.84± 0.38 | **89.33± 0.34** | 85.75± 0.53 | **89.28± 0.44** | 86.67± 0.53 | **89.76± 0.37** |
| | No | 500, 10 | 89.03±0.46 | **90.12±0.26** | **87.13±0.44** | 87.15±0.32 | **87.02±0.08** | 87.18±0.05 | 87.78±0.08 | **88.44±0.08** |
| | | 100, 10 | 89.88± 0.32 | **91.37± 0.16** | **90.62±0.16** | 90.83± 0.22 | 87.38±0.12 | **88.24±0.16** | 90.27±0.08 | **91.10±0.06** |
| | | 10, 10 | 89.37± 0.83 | **90.36±0.61** | 89.61±0.88 | **90.22±0.78** | **88.39±0.86** | 89.4±0.84 | **88.78±0.83** | 89.88±0.75 |
| **Quantity Skew** | | | | | | | | | | |
| CIFAR10 ($\beta = 0.5$) | - | 100, 10 | 79.36±0.32 | **80.61±0.24** | 80.12±0.38 | 80.88±0.18 | 79.24±0.38 | **82.95±0.41** | 79.81±0.32 | **81.62±0.48** |
| | | 10, 10 | 84.78±0.44 | **86.28± 0.38** | 83.11±0.55 | **84.42±0.46** | 85.92±0.16 | **86.58±0.28** | 84.98±0.15 | **86.47±0.29** |

Table 3: Real-world evaluation on FEMNIST, feature skew and mixed (feature and label) skew on FedCMNIST and FedRotMNIST, and Quantity skew on CIFAR 10. Total number of clients and number of clients sampled for participation in each communication round is as given in the corresponding row. Average best test performance of the algorithms and their GMA versions are reported below. The best average result among AVG and GMA having atleast 1% higher than the other algorithm is shown in bold.

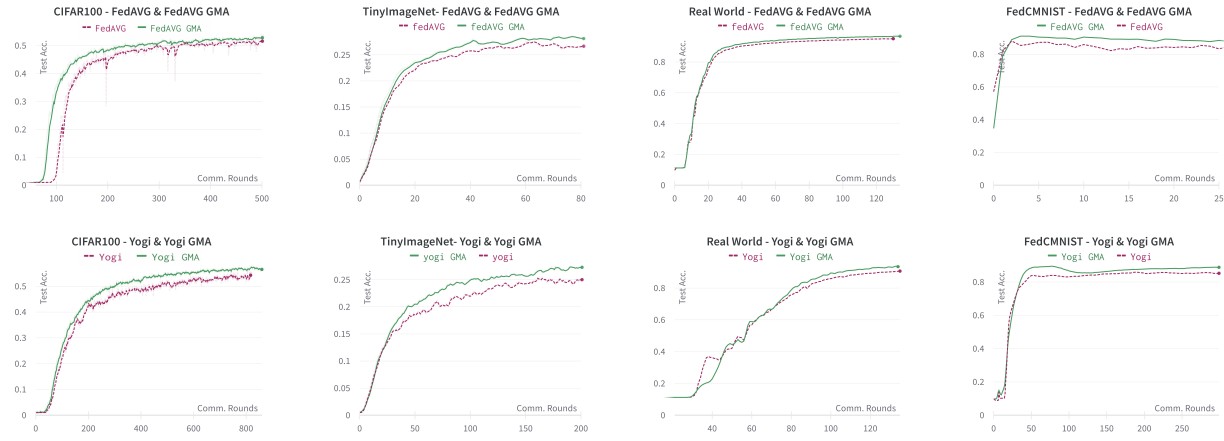

Figure 2: Test accuracy vs. communication rounds of non-adaptive (FedAVG) and adaptive (FedYogi) federated algorithms and their GMA versions on in-distribution CIFAR 100 and TinyImageNet, real-world FEMNIST, and mixed skew FedCMNIST datasets.

## 5.3 Out-of-Distribution Evaluation

A more complex OOD test can be implemented to better understand the performance of gradient masking in federated learning. For this we induce unique spurious mechanisms or features in the clients (including test) besides the class label based heterogeneity. The global model would be tested on a dataset having a spurious mechanism that was not present in any of the train clients, while the spurious mechanism at each train client is unique to itself. This ensures that all clients (including test clients) are out-of-distribution to each other. We combine this with label skew that is similar to the non-i.i.d heterogeneity in McMahan et al. (2017). FedCMNIST Francis et al. (2021), a federated multiclass version of CMNIST Arjovsky et al. (2020) with multiple color-label correlations was used for this. The invariant mechanism here is the digit. There also exists a spurious mechanism marked by a color given to the numbers. The color is digit specific to induce correlation to the label. The performance of various algorithms and their gradient masked averaging counterparts on these datasets is given in Table 3. It is to be noted that in almost all cases gradient masking outperforms naive averaging. Figure 3 a. shows train and test curves of the algorithms and their gradient masked alternatives on non-i.i.d distribution across 10 clients with full participation on FedCMNIST. It is to

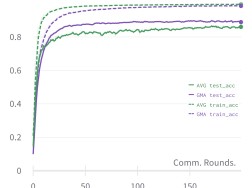 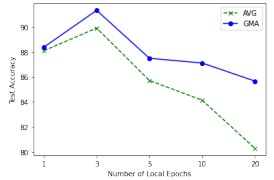 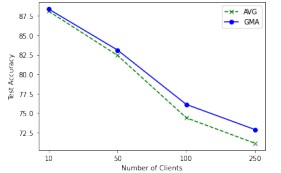 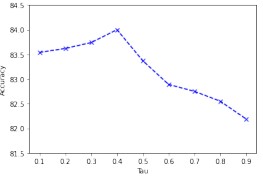

a) Acc. vs. Comm. Round    b) Test Acc. vs. Local Epochs    c) Test Acc. vs. # Clients    d) Acc. vs. $\tau$

Figure 3: Ablations: a) Train and test accuracies of FedAVG on FedCMNIST with 10 clients and full participation. b) Test accuracies of FedAVG and GMA on 10 client FMNIST with full participation is recorded against increasing number of local epochs. c) Test accuracies against number of clients in the federated network while all clients participate in training in each round (full participation). d) Test accuracy against $\tau \in \{0, 1\}$ on label skewed CIFAR10 distributed across 10 clients with full participation.

be noted that although GMA train accuracies are less than or equal to that of naive averaging, the GMA test accuracies are higher. This indicates that gradient masked versions generalize better than the naive versions.

## 5.4 Quantity Skew

In a practical federated setting the quantity of data available for update at each client varies drastically. Depending on connectivity, processing power, user behaviour, and various other factors, the number of data samples generated at each client can differ from zero or one data point to an extremely large number. To simulate this quantity imbalance, we have used a Dirichlet distribution based quantity skew with $\beta = 0.5$ as in Li et al. (2021a) on CIFAR-10 across 100 and 10 clients with 10 clients participating in each communication round. The Dirichlet sampling is independent of the labels or features of the dataset. The test data contains samples from all classes. Table 3 shows that GMA outperforms AVG under quantity imbalances.

## 5.5 Ablations and Analysis

**Ablations** Using the non-iid distributed FMNIST data we study how the performance is affected as the number of clients grows while all clients participate in training in each round ($N = 10, 50, 100, 250$) (shown in Figure 3 c.) and the number of local epochs increases ($E = 1, 3, 5, 10, 20$) (shown in Figure 3 b.).

Table 4: Test accuracies of paticipating and non-participating clients of a single FedAVG round with and without GMA on FMNIST distributed across 10 cients. Participating clients of a round are those that are randomly sampled to contribute to training in that round and the remaining are the non-participating clients.

| Trial | Participating | | Non-Participating | |
|---|---|---|---|---|
| | AVG | GMA | AVG | GMA |
| 1 | 84.34 | 85.67 | 81.63 | 85.39 |
| 2 | 87.22 | 89.44 | 72.38 | 73.04 |
| 3 | 89.34 | 92.11 | 71.21 | 77.18 |
| Mean: | 86.97 | **89.07** | 75.07 | **78.53** |
| Relative Improv.: | | 2.4% | | 4.6% |

In the former, the data is distributed across 10 clients with varying number of local epochs and in the latter, the data is distributed across a varying number clients with local epochs =1. In both cases all clients participate in training in each communication round. It can be observed from various datasets and settings that gradient masking increasingly outperforms naive averaging in complex datasets and sampled settings. This validates the enhanced invulnerability of gradient masking to the bias that could be induced by one or more clients in the network.

**Convex Objective** To further understand the performance of gradient masking in the convex setting, we experiment with MNIST and FedAVG on both i.i.d and non-i.i.d data distributions and it was observed that GMA outperforms naive averaging in the non-iid setting. A logistic regression model with SGD with momentum optimizer was used at the clients for these experiments. When data distribution was i.i.d, GMA was converging to an average (over last 10 communication rounds) of 92.5% test accuracy while naive averaging obtains to

92.4%. Furthermore, when the data distribution across clients was non-i.i.d, the enhancement in performance was more significant with gradient masking. While naive averaging was converging to 87.0% test and 92.0% train, while GMA reached 88.5% test and 92.2% train. Further demonstrating GMA can generalized better in the non-iid case.

**Effect on non-participating clients**   GMA tends to improve performance gains in the relevant setting with lower client participation. We can gain insight into this by considering an individual FL round where the non-participating clients data are out-of-distribution with respect to the obtained model, as their data is not used to update the global model in that round. Motivated by this observation we further study the behavior of a single FedAVG round with and without GMA on the FMNIST dataset using 3 local epochs on 10 clients with 5 clients participating in each round. The results are shown in Table 5 where we report the accuracy over participating and non-participating clients. We observe that GMA improves performance in both cases but the relative performance change on non-participating clients is higher (4.6%) versus 2.4%). This suggests the performance gain in lower client participation is stronger due to improvement on non-participating client performance.

## 6   Conclusion

We proposed a new aggregation scheme applicable to a wide variety of federated learning algorithms. The proposed method, gradient masking enhances generalization performance of the global model in FL by focusing on learning the invariances across clients. The simple masking outperforms their naive averaging versions across a variety of algorithms and datasets. Our theoretical analysis shows the convergence of the proposed masking algorithm and the stability of the proposed mask. Future directions include exploration of masks incorporating magnitude and other methods to better capture the invariances, thus leading to better generalization at the global model.

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

# A   Appendix

## A. Drifting Bound of k

Below is the proof of Lemma 4.7:

*Proof.* All the core idea of this proof is to observe that for the batch of data $z_{k,n}^t$:

$$\|\Delta_t + K\eta_l \nabla f(w_t)\|^2 = \eta_l^2 \|\sum_{k=1}^{K}\sum_{n=1}^{N} \frac{s_n}{\sum_{n=1}^{N} s_n}\left(\nabla f_n(w_{t,k}^n; z_{k,n}^t) - \nabla F_n(w_t)\right)\|^2 \tag{9}$$

$$\leq \eta_l^2 K \sum_{k=1}^{K} \|\sum_{n=1}^{N} \frac{s_n}{\sum_{n=1}^{N} s_n}\left(\nabla f_n(w_{t,k}^n; z_{k,n}^t) - \nabla F_n(w_t)\right)\|^2 \tag{10}$$

$$\leq \eta_l^2 K \sum_{k=1}^{K}\sum_{n=1}^{N} \frac{s_n}{\sum_{n=1}^{N} s_n} \|\nabla f_n(w_{t,k}^n; z_{k,n}^t) - \nabla F_n(w_t)\|^2 \tag{11}$$

$$\tag{12}$$

We also observe that:

$$\mathbb{E}[\Delta_t + K\eta_l \nabla f(w_t)] = \sum_{k=1}^{K}\sum_{n=1}^{N} \frac{s_n}{\sum_{n=1}^{N} s_n}\left(\nabla F_n(w_{t,k}^n) - \nabla F_n(w_t)\right)$$

Now, we obtain that by taking the expectation and $L$-smoothness that:

$$\mathbb{E}[\|\Delta_t + K\eta_l \nabla f(w_t)\|^2] \leq \eta_l^2 K \sum_{k=1}^{K}\sum_{n=1}^{N} \frac{s_n}{\sum_{n=1}^{N} s_n}(\|F_n(w_{t,k}^n) - \nabla F_n(w_t)\|^2 + \sum_{j=1}^{d}\sigma_{g,j}^2) \tag{13}$$

$$\leq \eta_l^2 K \sum_{k=1}^{K}\sum_{n=1}^{N} \frac{s_n}{\sum_{n=1}^{N} s_n}(L^2\|w_{t,k}^n - w_t\|^2 + \sum_{j=1}^{d}\sigma_{g,j}^2) \tag{14}$$

and we can use Lemma 4.6 and we obtain the conclusion (using $1 \leq K$) and a similar reasoning for the other part of the lemma.

$$\square$$

## B. Algorithms

This section gives the pseudocode of the the proposed gradient masking on FedAVG, FedADAM, and FedYogi. GMA can be plugged into most federated learning algorithms easily. Algorithm 2 shows how to extend FedAVG to also include FedYogi and FedAdam and highlights how the masking interplays with this.

---

**Algorithm 2** Gradient Masked FedAVG McMahan et al. (2017), FedADAM , and FedYogi Reddi et al. (2021)

---

**Server Executes:**

  Initialize $w_0$

  **for** each server epoch, t = 1,2,3,... **do**

    Choose C clients at random

    **for** each client in C, n **do**

      $w_t^n = \text{ClientUpdate}(w_{t-1})$

      $\Delta_t^n = \frac{s_n}{\sum_{k \in C} s_k}(w_t^n - w_{t-1})$

    **end for**

    $\Delta_t = \sum_{n \in C} \Delta_t^n$

    $m_t = \tilde{m}_\tau(\{\Delta_t^n\}_{n \in C})$

    $z_t = \beta_1 z_{t-1} + (1 - \beta_1)\Delta_t$

    $v_t = v_{t-1} - (1 - \beta_2)\Delta_t^2 \text{sign}(v_{t-1} - \Delta_t^2)$

    $v_t = \beta_2 v_{t-1} + (1 - \beta_2)\Delta_t^2$

    $\Delta_t = \frac{z_t}{\sqrt{v_t} + e^{-3}}$

    $w_t = w_{t-1} + \eta_g * m_t \odot \Delta_t$

  **end for**

**ClientUpdate(w, n):**

  Initialize $w_0 = w$

  **for** each local client iteration, $k = 0, 1, 2, 3, .., K - 1$ **do**

    Sample $z \sim \mathcal{D}^n$

    $g_k = \nabla_w F_n(w_k; z)$

    $w_{k+1} = w_k - \eta_l \ g_k$

  **end for**

  **return** $w_K$ to server

---

## C. Additional Motivation

### C.1 Agreement and Homogeneity

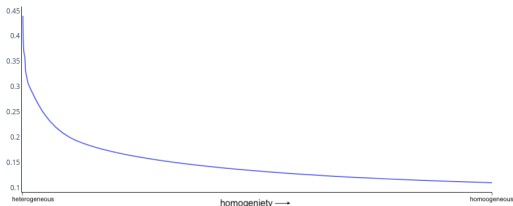

Figure 4: fraction of gradients having agreement < tau (0.4) across clients vs. homogeneity / heterogeneity od data distribution across clients.

We analyze the effect of data heterogeneity on the fraction of gradients that were less that $\tau = 0.4$. The gradients were adjusted to make update proportional to their agreement across clients. It was observed that with decreasing heterogeneity or increasing homogeneity, this fraction was reduced as shown in Figure 4. That is, with increasing homogeneity, the fraction of clients that were proportionately reduced decreased.

Alternatively, with increasing homogeneity, the number of gradients having agreement > tau increased. This indicates the relevance of the gradient agreement across clients in federated learning. The heterogeneity was induced using a Dirichlet distribution with $\alpha = 0.1$ representing heterogeneous case and $\alpha = 100$ representing homogeneous case. The experiments were based on label skewed MNIST.

## D.2 GMA vs. Binary Mask

According to Parascandolo et al. (2020), though the binary AND-Mask they proposed improves generalization performance, it is slower in convergence. The same was observed in a federated setting. The binary mask converges to a higher test accuracy than FedAVG and approximately equal to the proposed GMA. However, the number of communication rounds required to achieve it was higher when the data was non-iid as shown in Figure 5. In a federated setting, communication bottleneck is severe and improved convergence speed cannot be sacrificed for a minor improvement in generalization performance.

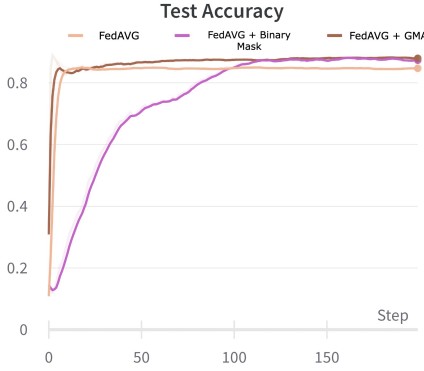

Figure 5: Comparison on FedAVG, proposed FedGMA, and the binary mask proposed by Parascandolo et al. (2020). The experiments were conducted on MNIST with label distribution skew.

## D.3 Effect on non-participating clients - 2 clients per communication round

Table 5: Test accuracies of paticipating and non-participating clients of a single FedAVG round with and without GMA on FMNIST distributed across 10 cients. Participating clients of a round are those that are randomly sampled to contribute to training in that round and the remaining clients are the non-participating clients.

| Trial | Participating | | Non-Participating | |
|---|---|---|---|---|
| | AVG | GMA | AVG | GMA |
| 1 | 89.73 | 87.72 | 71.61 | 74.34 |
| 2 | 90.70 | 85.57 | 71.01 | 76.56 |
| 3 | 88.5 | 86.73 | 72.09 | 76.43 |
| Mean: | 89.64 | 86.67 | 71.57 | 75.77 |
| Relative Improv.: | | -3.3% | | 5.8% |

In this section we extend effect on non-participating clients subsection of section 5.5 of the paper. We look at the effect of GMA on non-participating clients when only 20% of the clients participate in each communication round. In the table above, we consider 10 client with only 2 randomly chosen clients participating in each round. We observe that the relative performance improvement on non-participating clients is significant with GMA. We also note that performance of GMA on participating clients is lower than that of AVG. This is because GMA would update only if both clients agree on the sign. Due to this most parameters do not update in every round.

## D. Hyperparameters

### D.1 Effect of Tau

$\tau \in \{0, 1\}$ is a hyperparameter introduced to threshold agreement across clients in Equation 2. It marks the minimum agreement required to consider the gradient for aggregation. When $\tau = 0$ or negligible, the gradients of all parameters will have an agreement score greater than or equal to $\tau$. This makes all gradients consistent across clients. The agreement score would be over-written by 1 and the equation becomes equal to that of naive federated aggregation where all gradients are averaged with the same importance. This is equivalent to an underfit condition. The opposite overfit criterion can happen with a high $\tau$ value. In this case, no gradient would be considered dominant and all parameters updates would be diminished corresponding to their agreement score. The agreement score can be 1 when the data distribution across clients is an ideal i.i.d distribution where all gradients across clients would be along the same direction. But in practical scenario, such data distributions are rare in a federated setting. When agreement $= 1$, $w^k = w^{k-1} - \eta_g \, 1 \odot \Delta^k = w^k - \eta_g \Delta^k$; equivalent to naive federated aggregation. This implies that the naive fedrated aggregation is a case of the proposed gradient masked averaging.

Throughout the experiments we have maintained $\tau = 0.4$. This value was observed to give the best performance upon searching on $\{0, 1.0.2, 0.3, 0.4, 0.5, 0.6, 0.7, 0.8, 0.9\}$. $\tau = 0.4$ implies that the gradient being considered have a 40% excess or a total of 60% of the client gradients along the dominant direction. From our experiments on CIFAR10 distributed non-iid across 10 clients with full-participation, it was observed that when $\tau$ is low, the model underfits. The accuracy was best at $\tau = 0.4$ and on further increasing $\tau$, it was overfitting. This is visible from the test accuracy vs. $\tau$ plot in Figure 3 d. and test loss vs. $\tau$ plot in Figure 3d.

### D.2 Effect of Client Momentum

In contrast to the experiments in Reddi et al. (2021), we use an SGD optimizer with momentum ($\rho = 0.9$) at each client for all our experiments. This was primarily because of the increase in test accuracy observed during our experiments on FedAVG with and without momentum. However, the enhancements due to gradient masking was independent of the momentum induced at the client optimizer. In both cases (with and without momentum), gradient masking was outperforming naive averaging in most of the algorithms and datasets. Table 6 shows the performance of the algorithms and their GMA versions on non-i.i.d distributed FMNIST using an LeNet model. The client optimizers used in our experiments is a naive SGD optimizer with momentum parameter and it does not involve the correction parameter introduced in Xu et al. (2021).

### D.3 Effect of Group Norm

The test accuracies reported in paper corresponding to CIFAR-10 used a ResNet18 model with batch normalization layers replaced by group normalization(Wu & He, 2018) similar to the experiments in Reddi et al. (2021). Our initial experiments involved batch normalization as in the original ResNet and it was observed that the replacement of batch norm with group norm improved the test accuracies. Table 7 shows the comparison of the algorithms and their GMA versions on CIFAR-10 using ResNet model having batch normalization and group normalization layers. It is to be noted that irrespective of the normalization layer used, gradient masking was outperforming naive averaging across all algorithms and data distributions. This further validates the capabilities of the proposed GMA.

### D.4 Grid Search Range

For all our experiments a search for client learning rate ($\eta_l$) and global learning ($\eta_g$) rate across the grid specified below was conducted and the best performing learning rates of each algorithm was used for experiments that reported the test performances in the paper. The grid was fixed the same for all datasets and algorithms for easy experimentation.

Table 6: Performance of the algorithms and their GMA versions with and without momentum($\rho$) on non-i.i.d distributed FMNIST using an LeNet model on 10 clients with full-participation. Momentum improves performance of the algorithms. Irrespective of momentum, GMA outperforms AVG.

| Dataset (Model) | | FedAVG ($\rho = 0$) | | FedAVG ($\rho = 0.9$) | |
|---|---|---|---|---|---|
| | | AVG | GMA | AVG | GMA |
| MNIST | IID | **99.01** | 98.96 | 99.1 | **99.16** |
| (LeNet) | Non-IID | 98.43 | **98.55** | 98.87 | **98.9** |
| FMNIST | IID | **88.61** | 88.49 | 89.14 | **90.52** |
| (LeNet) | Non-IID | 86.95 | **87.8** | 88.1 | **88.38** |
| FEMNIST | IID | 98.8 | **98.92** | **99.7** | 99.68 |
| (LeNet) | Non-IID | 92.17 | **94.61** | 94.2 | **96.04** |
| CIFAR-10 | IID | 85.8 | **86.31** | 87.3 | **87.61** |
| (ResNet) | Non-IID | 81.1 | **82.28** | 83.25 | **83.95** |

Table 7: Average in-distribution test performance(%) over the last 10 communication rounds of FedAVG, FedProx, SCAFFOLD, FedAdam, FedYogi and their GMA versions on i.i.d and non-i.i.d distributions of CIFAR-10 on ResNet18 models using batch normalization and group normalization. The best result among AVG and GMA versions of each algorithm is shown in bold.

| Dataset (Model) | | FedAVG | | FedProx | | FedADAM | | FedYogi | |
|---|---|---|---|---|---|---|---|---|---|
| | | AVG | GMA | AVG | GMA | AVG | GMA | AVG | GMA |
| ResNet | IID | 87.11 | **87.42** | 87.2 | **87.52** | 77.32 | **80.65** | 78.78 | **80.55** |
| BatchNorm | Non-IID | 77.3 | **79.9** | 78.4 | **80.2** | 69.82 | **74.05** | 67.29 | **71.51** |
| ResNet | IID | 87.3 | **87.61** | 87.18 | **87.5** | 86.9 | **87.7** | 87.53 | **87.78** |
| GroupNorm | Non-IID | 83.25 | **83.66** | 83.87 | **84.4** | 83.53 | **84.84** | 83.17 | **84.55** |

$$\eta_l \in \{10^{-3}, 10^{-2}, 5.10^{-2}, 10^{-1}\}$$
$$\eta_g \in \{10^{-2}, 10^{-1}, 1, 1.5, 2\}$$
$$\tau \in \{0.1, 0.2, 0.3, 0.4, 0.5, 0.6, 0.7, 0.8, 0.9\}$$

The values for global learning rate and client learning rate was observed to vary across algorithms and datasets. However, it was observed that the same combination was giving the best performance for GMA and AVG versions of the same algorithm for the same dataset. This enables better comparison of naive averaging and gradient masked versions of algorithms better. For $\tau$, it was observed that $\tau = 0.4$ was giving the best result acriss algorithms and datasets. More details about hyperparameter $\tau$ is given in Appendix D.1.

## D.5 Increased Clients and Local Epochs

Table 8 and Table 9 shows average test accuracies corresponding to Figure 3 b and c. These report the average test performance over last 10 communication rounds on convergence of FedAVG and its GMA version across increasing number of selected clients and number of local epochs at each client per communication round.

Table 8: Average test performance values of non-i.i.d FMNIST on varying number of clients

| Number of Selected Clients | | 10 | 50 | 100 | 250 |
|---|---|---|---|---|---|
| FedAVG | AVG | 88.1 | 82.42 | 74.39 | 71.11 |
| | GMA | **88.38** | **83.11** | **76.11** | **72.87** |

Table 9: Average test performance values of non-i.i.d FMNIST with varying number of local client epochs per communication round

| Number of Local Epochs | | 3 | 5 | 10 | 20 |
|---|---|---|---|---|---|
| FedAVG | AVG | 89.91 | 85.71 | 84.13 | 80.27 |
| | GMA | **91.73** | **87.29** | **87.11** | **86.66** |

## D.6 Performance for same learning rates

Table 10 shows the in-distribution test performance of naive and gradient masked versions of FedAVG on non-iid FMNIST dataset. Performance corresponding to the entire range of hyperparameters mentioned in Appendix C.4 has been given here. It can be noted that across learning rates where the model learns, gradient masking outperforms naive averaging. This suggest the gains are highly robust to hyperparameter choices.

Table 10: Performance of FedAVG across a range of global learning rate ($\eta_g$) and client learning rate ($\eta_l$) on non-iid FMNIST distributed across 10 clients with full participation. It can be observed that GMA outperforms AVG in most of the cases where the algorithms learn and converge.

| $\eta_g$ | 0.001 | 0.01 | 0.05 | 0.1 | 1.0 | |
|---|---|---|---|---|---|---|
| 0.01 | 0.1 | 56.06 | 61.66 | 68.02 | 0.1 | AVG |
| | 0.1 | **57.72** | **65.13** | **72.56** | 0.1 | GMA |
| 0.1 | 56.93 | 73.66 | 83.39 | 85.22 | 0.1 | AVG |
| | **57.51** | **73.9** | **83.79** | **87.22** | 0.1 | GMA |
| 1.0 | 72.69 | 86.49 | 87.76 | 88.31 | 0.1 | AVG |
| | **73.34** | **87.0** | **88.14** | **88.4** | 0.1 | GMA |
| 1.5 | **77.11** | 86.29 | 87.82 | 86.96 | 0.1 | AVG |
| | 75.63 | **87.04** | **88.21** | **88.3** | 0.1 | GMA |
| 2.0 | 71.53 | 82.42 | 86.93 | 87.63 | 0.1 | AVG |
| | **77.59** | **86.9** | **87.6** | **88.1** | 0.1 | GMA |
| | 0.001 | 0.01 | 0.05 | 0.1 | 1.0 | |

$\eta_l$

### E. Convex Objective

For the experiments on convex objective we employed a logistic regression model on MNIST with FedAVG for global model approximation at the server. The model was created using a single linear layer in a perceptron model without any non-linear activation functions. A cross-entropy loss was used and the model was updates by an SGD optimizer with momentum at each client. Each client undergoes 1 update step before aggregation (gradient masked or naive avergaing) to approximate the global model. Experiments included i.i.d and non-i.i.d data distribution of data across clients with in-distribution test for global model evaluation. The number of clients selected at each round was set to 10 and the models were run until convergence. The various hyperparameters involved in the experiments are global model learning rate of 1.0, client model learning rate of 0.01, and $\tau = 0.4$.

### F. Membership Inerence Attack

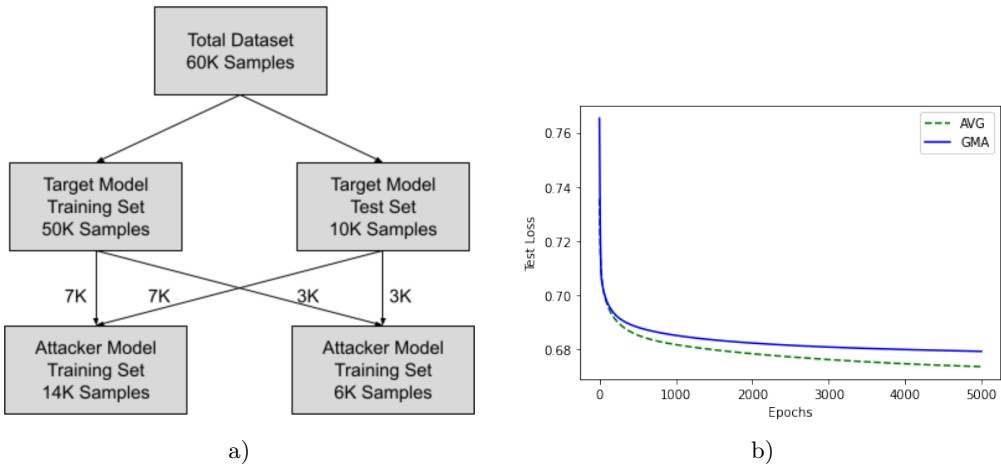

Figure 6: (a) Data split and creation for attacker model (b) Test loss vs. epochs of the logistic regression attacker model.

Most machine learning models tends to overfit on their training data and such models are susceptible to membership inference attacks that can accurately predict whether a data sample was present in the training set of the model given the model output logits Tople et al. (2020). This is a major privacy breach and it can simulated by using a black-box adversarial attacker model.

The attacker model we have employed is a binary logistic regression model with binary cross-entropy loss. The input to this attacker model is the logits of the converged gradient masked averaging model and naive averaging model. The attacker model is supposed to identify whether the input of the global model corresponding to the logit given was present in the global model's training set or not. For our experiments, we used CIFAR-10 and ResNet models. Firstly, the GMA and AVG models were trained and tested. For each model, the logits corresponding to the train and test set data and their labels (whether train data or test data) were stored. The data is split as shown in Figure 7 a Tople et al. (2020) and the attacker model is trained for 5000 rounds. The accuracy is as reported in the paper and loss as shown in Figure 7 b . A lower attack accuracy of gradient masked implies that GMA has better immunity to membership inference attacks than naive averaging global models.

This experiment is based on Tople et al. (2020) which suggests that algorithms focusing on learning the causal mechanisms provide stronger privacy guarantees in certain cases, for example they can be more robust to membership inference attacks and model inversion attacks. Based on our experiments we observe that the

attack accuracy with respect to the GMA model is 55% while that of naive averaging model is 57%. This suggests that gradient masking can potentially enhance robustness to membership inference attacks.

### G. Mask Stability

Below we establish a concentration bound, to determine the likelihood of applying the masking operation to a coordinate

**Proposition A.1** (Mask stability). *Denote $\delta, \delta^n$ the r.v. corresponding respectively to a coordinate of $\Delta, \Delta^n$. Furthermore consider $\tilde{\delta}$ the r.v. for each coordinate of $\tilde{\Delta}$, where $\tilde{\Delta} = m_t \odot \Delta$. Assume that $\delta^n$ is $\sigma$-subGaussian, that the $\delta_n$ are mutually independent and write $\mu^n = \mathbb{E}[\delta^n]$. If $\frac{1}{N}\mathbf{card}(\{n|\mu^n > 0\}) > \tau$, then, with probability $1 - \mathcal{O}\left(e^{-\frac{(\inf_{\mu_n > 0} \mu_n)^2}{\sigma^2}}\right)$, we obtain $\tilde{\delta} = \delta$.*

*Proof.* We show a lower bound on $\delta^n$. With probability $1 - e^{-\frac{t^2}{\sigma^2}}$, we get $|\delta^n - \mu^n| < t$. Let's thus pick $\tilde{t} = \inf_{\{\mu^n > 0\}} \frac{\mu^n}{2}$. By considering the intersection of those events, it implies that with probability at least $1 - e^{-\frac{\tilde{t}^2}{\sigma^2}}$, $\delta^n > \mu^n - \frac{\mu^n}{2} = \frac{1}{2}\mu^n > 0$. Consequently, the mask is equal to 1 and $\delta^n = \tilde{\delta}^n$. $\qquad\square$

