# OpenReview forum: "Gradient Masked Averaging for Federated Learning"
_TMLR — Accepted by TMLR_

### Review · Reviewer_ffYZ · 2023-06-25

**Summary Of Contributions:**

The paper proposes gradient maks averaging (GMA) for Federated Learning (FL). Specifically, the method computes an element-wise soft mask, which has values between 0 and 1, and applies it to the aggregated gradients from all clients before applying the update to the global model.  The mask is calculated based on the principle of agreement among clients, inspired by related works in out-of-distribution (OOD) generalization. The paper applies the method to several existing FL algorithms and demonstrates better performance than the original algorithms on various FL settings including artificial IID and Non-IID datasets, real-world FL datasets, and OOD datasets with spurious features.

**Audience:**

Yes

**Broader Impact Concerns:**

The reviewer does not see potential ethical implications of this submission.

**Claims And Evidence:**

Yes

**Requested Changes:**

* The authors should provide more clear writing of the theory component and discuss the motivations and implications of theoretical results. It is ok to have assumptions but still, it is not clear what the convergence results indicate under the current assumptions. Also, proposition 4.9 is very confusing in its current state. (**critical**)

* Please fix the typos mentioned in the previous section. (**improving presentation**)

**Strengths And Weaknesses:**

**Strength**
* The method of masking is well-motivated and novel in the context of FL. Specifically, the idea of using masking to encourage consensus and avoid learning from spurious features has been explored in OOD generalization literature. This paper treats each client as a separate environment whereas each sample is treated as an individual environment in OOD generalization. Furthermore, establishing a high-level connection between OOD generation and federated learning is also an interesting aspect as both fields share some common underlying challenges such as spurious correlations in the form of heterogeneity.

* The experiments are extensive. The paper provides experiments on many different FL settings including common simulated IID and Non-IID splits and real-world datasets. Moreover, the paper also provides results on OOD experiments where OOD data have different spurious mechanisms. It is intriguing to see that masking can improve both federated performance and also OOD performance in a federated setting.

**Weakness**
* The theory is not specifically enlightening. The main convergence result (proposition 4.8) relies on an assumption of a randomly sampled mask instead of the real mechanism proposed. While the theoretical difficulty of analyzing the original algorithm can be understood, this nevertheless weakens the contribution.
* The writing of the theory section is also confusing. There is little discussion on theoretical results and their implications. For example, it is not clear why Proposition 4.9 (mask stability) is important and what it indicates about the masking method.

**Minor**
* At the beginning of 3.1, the paper wrote "$f_n$ is an estimate of the corresponding gradient". Should $f_n$ here denote the empirical loss/objective function given sample data?
* In equation 1, $\eta_g\Delta_t^n$ should be  $\eta_g\Delta_t$.
* At the beginning of 3.2, the notation $m_\tau(\delta)$ is not clarified.

---

> ### Author Response · Authors · 2023-07-03
> **Thank you for the review**
>
> We thank the reviewer for their thorough reading of our paper and comments which allow us to improve the quality. We have made some revisions to clarify the notation and  improve the theoretical section.
>
> Regarding Proposition 4.8. We agree that this theoretical result does not provide a full characterization of all the desired properties. We emphasize however the primary contribution of our paper is empirical and methodological. This theoretical analysis serves as a sanity check, providing insights that the process of masking is unlikely to significantly impede convergence towards a particular local minimum. However, it is important to note that this theorem  alone does not explain the observed strong performance. Instead, it indicates that there are no inherent obstacles preventing the algorithm from converging to a local minimum. Therefore, this simplification of the problem setting serves as a valuable starting point for further exploration and refinement of the masking approach, allowing for the investigation of more realistic and complex scenarios.
>
> We have added clarifications to this effect at the end of Section 4.
>
> Concerning Proposition 4.9, our initial aim was to establish a simple probabilistic condition, based on a concentration bound, to determine the likelihood of applying the masking operation to a coordinate. However, after carefully reviewing the comments of the reviewer and reevaluating this theorem, we have come to the conclusion that its contribution is limited to its nature as a concentration bound and is not particularly insightful. Furthermore, incorporating it into the main body of the paper may impede readability and divert attention from the primary contributions of our work. Therefore, we have made the decision to relocate this section to the appendix, enabling us to concentrate more effectively on the central aspects of our study.
>
> We thank the reviewer for pointing out the typos, we have corrected them.

---

### Review · Reviewer_54rX · 2023-07-11

**Summary Of Contributions:**

The paper proposes an aggregation mechanism, Gradient Masked Averaging (GMA),  for federated learning. Compared to conventional FL that simply averages over local parameters, the proposed method is more robust to out-of-distribution data and can be used as a component for many existing federated learning algorithms. Under GMA, each client after receiving global model performs multiple local updates, the sign of each update component is then used to compute element-wise agreement score. A soft-masking operation is applied subsequently to determine the final global update.

**Audience:**

Yes

**Claims And Evidence:**

No

**Requested Changes:**

1. The authors mentioned that due to highly-correlated masked gradients, the convergence analysis of FedAvg with GMA is hard to obtain. Can you elaborate more on this? Compared to conventional FedAvg, how is the convergence rate affected when applying GMA? Under heterogeneous clients, isn’t the assumption that binary mask follows independent Bernoulli distribution easily violated?
2. How do we know theoretically GMA improevs generalization to out-of-distribution data?

**Strengths And Weaknesses:**

Strengths
1. The proposed method can be applied directly to many existing federated learning algorithm, by serving as a drop-in robust alternative to naive averaging of parameters.
2. The proposed GMA is a simple mechanism and is easy to be adopted. Using FedADAM as an example, the authors have shown that under the assumption that binary mask follows independent Bernoulli distribution, the modified FedADAM with GMA still converges.

Weaknesses
The authors claimed that the proposed GMA, when applied to existing FL algorithms, can generalize better to out-of-distribution data without sacrificing the convergence rate. However,  there is no rigorous support for such a statement. Theoretically, the authors only provide the convergence analysis for a special case: FedAvg with GMA under the assumption that binary mask follows independent Bernoulli. It is unclear whether GMA can really help generalization to out-of-distribution data. Empirically, the paper only applies GMA to 4 existing FL algorithms: FedAvg, FedProx, FedAdam, FedYogi, but the improvement over naive averaging mechanism doesn’t seem to be significant. To fully evaluate GMA, many other baselines should be considered.

---

> ### Author Response · Authors · 2023-08-25
> **Response**
>
> We thank the reviewer for their thorough reading of our paper and comments which allow us to improve the quality. Below we address the individual concerns. We want to emphasize globally that we agree with the limitations of the theory related to the work that the reviewer has noted and also these limitations are already clearly stated in the paper as detailed below.
>
> *elaboration on “highly-correlated” In our analysis, we need a lower-bound on the norm of the masked gradients (see Prop 4.8, and the inequality $\alpha \Vert \nabla f(w_t)\Vert^2\leq \mathbb{E}\Vert m_t\odot\nabla f(w_t)\Vert^2$). Since the masks depend on the gradient, we need to formulate an assumption to obtain the corresponding lower-bounds on the norm of the masked gradients. We changed the formulation “highly correlated” at top of Sec 4 to say without additional assumptions to lower-bound the norm of our masked gradients, it is difficult to derive a proof.
> *Comparison of rate of GMA to FedAVG* Our results show the rate is similar, up to a multiplicative constant which depends on the mask: this is expected as our proof scheme follows very similar ideas from FedAvg.
> *Validity of Assumption 4.5* As noted, this assumption is there to lead to the inequality:  $\alpha \Vert \nabla f(w_t)\Vert^2\leq \mathbb{E}\Vert m_t\odot\nabla f(w_t)\Vert^2$. While we agree that is a simplification that can be violated in some practical cases, any other assumptions allowing us to get this inequality would be amenable to our work.
> At the bottom of Section 4 it is already stated the Assumption 4.5 could be violated in some practical cases, but that the current convergence properties have to be understood as a sanity check.
>
>
> “How do we know theoretically GMA improve generalization to out-of-distribution data?”
> Our work is largely empirical and motivated/inspired by Pascanado et al with respect to the OOD generalization. Pascanado et al work also only shows limited theoretical guarantees. Specifically, the notion of ILC is used which does not guarantee out of distribution generalization, and furthermore the only theoretical results are shown on full batch gradient descent. Indeed many works focusing only on OOD generalization methods do not provide strong theoretical guarantees.
> Our paper does not claim at any point theoretical guarantees on OOD generalization as stated in the contribution section and the claims regarding improvements of GMA on OOD generalization are supported by experiments.

---

### Review · Reviewer_Q7tq · 2023-07-28

**Summary Of Contributions:**

This paper provides a new aggregation scheme that can be used by various federated learning algorithms. The proposed method hinges upon soft masking of the local gradients prior to updating the global model. The authors then conduct extensive experimentation showing the efficacy of their approach and its impact in various settings.

**Audience:**

Yes

**Broader Impact Concerns:**

There are no broader impact concerns.

**Claims And Evidence:**

Yes

**Requested Changes:**

1) At Algorithm 1, please replace $n_s$ with $s_n$.
2) At Assumption 4.3, please make that the expectation sign is concerning a valid norm; currently, the norm is not written correctly.
3) What do the lines with light colors mean in Figure 5 in Section D.2 (in the appendix)?

**Strengths And Weaknesses:**

Strengths:
 * The paper suggests a novel idea of using a "soft" gradient masking.
 * A binary masking which can be thought of as a predecessor to this method does underperform against the proposed aggregation scheme which shows the need for softer masks.
 * The theory stated in the paper is simple to understand and for the most part, beautifully put.
 * From a practical point of view, the proposed method outperforms the AVG aggeration method which can be considered a naive and widely used aggeration method.
 * The authors have conducted extensive experimentation highlighting and showing the pros and cons of their proposed method.

Weaknesses:
  * The authors propose the use of soft masks as opposed to using binary masks. However, their theoretical guarantees are done with respect to binary masks, via the use of Bernoulli variables. This is understandable since analyzing the convergence associated with softer masks is hard to obtain.
  * There are some typos across the paper; see the following section.

---

> ### Author Response · Authors · 2023-08-25
> **Response**
>
> We thank the reviewer for their thorough reading of our paper and comments which allow us to improve the quality. We have corrected the requests 1 & 2.  The lines in the light color in Figure 5 in Section D.2, are the original line plots. The darker lines represents a smoothened version of the same line.  With respect to the weakness of the analysis being limited to binary masks, we agree that a full theory involving the soft mask would be an interesting future direction.

---

### Decision · Action_Editors · 2023-09-18

**Recommendation:** Accept as is

**Comment:**

Main comments by the reviewers:
- the idea is "well-motivated" [ffYZ], "novel" [ffYZ, D7tq] and simple and "easy to be adopted" [54rX]. It "can be applied to many existing federated learning algorithms" [54rX].
- the theory is clear and "beautifully put" according to [D7tq] but on the other hand, the "writing of the theory section is also confusing" [ffYZ].
- the theory is limited: "the authors only provide the convergence analysis for a special case" [54rX], that is, i.i.d. Bernoulli masks. Thus, they do not completely explain why the method works. "While the theoretical difficulty of analyzing the original algorithm can be understood, this nevertheless weakens the contribution" [ffYZ]. This is acknowledged by the authors. They highlight the difficulty of studying the general case.
- the experiments are extensive [ffYZ] and honest, as they show "the pros and cons of the proposed method" [D7tq]. I will note a claim by [54rX]: "the improvement over naive averaging mechanism doesn’t seem to be significant", that seems to be in contradiction with most of the experimental results (if we interpret 'significant' as 'statistically significant') and with the claims of the other reviewers: " the proposed method outperforms the AVG aggeration method" [D7tq], "it is intriguing to see that masking can improve both federated performance and also OOD performance in a federated setting" [ffYZ].

The few typos pointed out by [ffYZ, D7tq] were fixed by the authors. They also slightly re-organized the theoretical part to address the comment on clarity by [ffYZ].

As a conclusion, the revised version of the paper proposes a new method that is easy to use in practice, well supported by experiments. They proposed a preliminary theory that, despite its clarity, does not explain why the method works in its general form. For [D7tq], the practical advantages are enough to support fully publication. [ffYZ] has more reservation because of the theory. Still, this reviewer is also leaning towards acceptance. I assume that he/she considers that the experimental results are enough to support the publication of the paper. On the other hand, [54rX] is leaning towards rejection. However, he/she did not reply to the authors rebuttal, and did not provide any explanation motivating this decision. I can only infer that for this reviewer, the experimental results do not outweigh the lack of theory. As both reviewers and myself disagree with [54rX]'s evaluation of the empirical result, I feel safe enough to recommend to accept this paper (I also note that, would [54rX] had strong objections to the acceptance of the paper, he/she would have stated it clearly anyway). Overall, it's a promising method well supported by a strong experimental analysis. A full theory seems to be quite challenging, but we can hope this problem will receive attention in the coming months.

**Audience:**

Federated learning is a large and growing community in ML. Moreover, [reviewer ffYZ] points out that the connection to the out-of-distribution (OOD) generalization literature makes the paper relevant beyond the federated learning community.

**Claims And Evidence:**

The authors study federated learning. More precisely, they focus on aggregating gradients computed on heterogeneous datasets. Obviously, a naive aggregation might be a bad idea in case some datasets are moire noisy, or corrupted: they might lead to very large gradients, that will dominate the average. The authors propose a new soft-mask method to weight the aggregation in a way to get consesus among the gradients. One of the striking advantages of this method is that it can be applied to virtually any existing federated learning algorithm. They provide both a rigorous theoretical analysis (in a limited special case) and exhaustive numerical experiments to support their method.